# Sustainable Careers of Teachers of Languages Other than English (LOTEs) for Sustainable Multilingualism in Chinese Universities

**Yadong Guo [1], Helena Sit [2,\*] and Min Bao [3,\*]**

1   School of Foreign Languages, Tongji University, Shanghai 200092, China; guoyadong127@tongji.edu.cn
2   School of Education, The University of Newcastle, Newcastle, NSW 2308, Australia
3   School of Foreign Languages, Southeast University, Nanjing 211189, China
\*   Correspondence: helena.sit@newcastle.edu.au (H.S.); ellen_bm@126.com (M.B.);
    Tel.: +61-4-0677-5986 (H.S.); +86-138-5166-8370 (M.B.)

**Abstract:** This paper explores Chinese universities' policies related to the research performance review of language other than English (LOTE) teachers with respect to promotion. Drawing on a variety of data including policy documents and interviews with 32 individual LOTE teachers from 16 universities, we identified that Chinese universities have unreasonable expectations in terms of research publications and research funding for language teachers, including LOTE teachers, which make their career prospects unsustainable. We also evaluated the contextual realities for LOTE teachers regarding academic publication and research funding, and identified a widespread feeling of anxiety and stress among LOTE teachers. Though LOTE teachers are committed to undertaking various efforts to overcome challenges in their research performance review for promotion, we call on university management and policy makers to provide additional support to LOTE teachers, so that they can develop sustainable careers and universities, including Chinese universities, will be able to rely on sustainable multilingualism.

**Keywords:** new managerialist; sustainable career prospect; language teachers in China; languages other than English (LOTE)

---

## 1. Introduction

The decline in the instruction of languages other than English (LOTE) in universities has created uncertainties for LOTE teachers globally [1,2]. In particular, the continuing decrease in the number of secondary school students taking modern languages has caused related university language programs to downsize in Europe [2,3]. The rise of English as a global lingua franca has also motivated universities in Asia to invest resources in developing English language programs, at the cost of other European language programs which are unable to attract sufficient numbers of students [1]. Consequently, LOTE teachers in these universities have been affected by low enrolment figures in related language programs and may have witnessed an increasing number of positions being cut and be facing increasingly insecure and unsustainable career prospects.

In contrast, LOTE teachers in Chinese universities have experienced more promising career prospects in the last few years due to the government's investment in LOTE-related programs. Many Chinese universities have been asked to start LOTE instruction and LOTE degree programs to satisfy the perceived need for graduates with LOTE skills in the context of the Chinese government's commitment to the 'Belt and Road' initiative, a new strategic blueprint for engagement with economic and social development in countries on China's historical trading routes [4–6]. Han et al. [6] reported 1417 LOTE programs in 583 universities across China in 2017. This growth in LOTE programs has made it

challenging for Chinese universities to recruit and retain LOTE teachers; Han et al. [6] noted that many LOTE programs were understaffed and LOTE teachers had junior academic ranks, often because they had only just completed their graduate or perhaps even undergraduate studies before commencing their employment. In other words, Chinese universities may have a large number of vacancies for LOTE teachers, but it remains questionable whether these newly appointed LOTE teachers will survive their institutions' increasingly challenging personnel evaluation procedures and be able to achieve sustainable career prospects in Chinese universities.

The question is also particularly pertinent as many Chinese universities, like their counterparts elsewhere in the world, are increasingly using new managerialist practices, aligned with "private sector practices and concerns, notably efficiency, effectiveness and excellence" [7] (p. 1668), to evaluate the work of university academics with a particular focus on their research performance. Therefore, it is important for us to explore whether the current research performance evaluation system in Chinese universities can facilitate LOTE teachers to have sustainable careers. To that end, this study addressed the following research questions:

1.　What are LOTE teachers expected to achieve in terms of research in order to facilitate promotion in Chinese universities?
2.　How sustainable is the current performance evaluation system in Chinese universities in relation to LOTE teachers?

The notion of sustainable careers refers to "sequences of career experiences reflected through a variety of patterns of continuity over time, thereby crossing several social spaces, characterized by individual agency, herewith providing meaning to the individual" [8] (p. 7). It needs to be explored and understood in terms of "the person, the context and time" [9] (p. 1). For this reason, we will first situate our inquiry in the context of rising managerialism or new managerialism in higher education, before we present a picture of the conditions in Chinese universities that will help us to appreciate the challenges faced by LOTE teachers during their career development.

## 2. Managerialism and Chinese Universities

The increased adoption of managerialism or new managerialism ideas and practices in university governance has profoundly changed the ways in which universities operate and how university academics work. It has been argued that universities need to adopt new methods to measure and assess scientific research outputs so that universities can "optimize our contributions to society and the world" [10] (p. 1). Researchers in contexts such as Europe have been exploring new measures such as next generation metrics to support early-career researchers' career development and ensure sustainability in academic career development [11]. The next generation metrics value individual researchers' "peer-reviewed publications in open access journals" and encourage universities to take up the challenge of contributing "to ecological, social and economic sustainability' in their operations" [11] (p. 12). Nevertheless, in most universities at present, university management at different levels still use "targets" to monitor, control, and evaluate operations to ensure "efficiency and effectiveness" [12] (p. 11). Consequently, university academics are expected to achieve outcomes measured by types of key performance indicators such as their academic publications and research grants [13–17]. In many universities, academics need to publish a certain number of publications and successfully apply for a certain amount of research funding in order for their positions to remain secure. Although such management practices have been challenged by research evidence, very few studies have examined how possible it is for individual academics to maintain sustainable career prospects, and how helpful these management practices are in facilitating them to pursue career success.

It must be noted that Chinese universities have quite different conditions for operation from their counterparts in contexts such as Australia, the UK, and the US. In many of these English-speaking countries, universities' operations are supported by a variety of revenue-generating activities (e.g., enrolling international fee-paying students) supplemented by diminishing income from

governments. In China, most universities are publicly funded institutions, benefiting from steady financial support from government sources at different levels. Although they may be less financially vulnerable than their counterparts in other parts of the world, their operations are closely monitored and supervised by the government. Because the Chinese government is driven by the aim that Chinese universities should be recognized as world class institutions, Chinese universities have also begun to adopt managerialist practices to improve their performance, in particular their research productivity, so that they can be favorably ranked in international university ranking exercises. A series of government initiatives such as 'Project 211' and 'Scheme 985' have injected funding into a selection of Chinese universities, and following the launch of these initiatives university academics found themselves increasingly accountable in terms of their performance evaluation (e.g., their research output).

The Chinese government invested 2.2 billion USD in over 100 universities included in Project 211 up to the year 2000 [18]. The government provided 14 billion RMB to 34 universities in the first phase of Scheme 985, and another 18.9 billion RMB to 39 universities in the Scheme's second phase [19,20]. The newly-launched initiative to establish 'first-class' universities and academic disciplines obliges the government to invest heavily in a few elite universities and a highly selective list of programs in other universities, so that these universities and programs will be ranked as international leaders in major exercises including the QS World University Rankings.

Research performance is one of the key performance indicators, using metrics such as numbers of citations and publications, with the consequence that this is also one of the few indicators where university management may be able to see measurable visible improvements with appropriate investment of efforts and funding. Therefore, Chinese universities that are already included in this 'Double First-class' initiative, or which may be aspiring to join the initiative, have placed even more emphasis on raising university academics' research output so that the universities can "enhance their capacity for fundamental research and compete in cutting edge research and even lead the research in these disciplinary areas" [21] (p. 1).

Consequently, academics in these universities are increasingly pressurized to publish more research in quality publication outlets [22,23]. In the academic departments where LOTE teachers are likely to work, academic research performance is already considered weaker than in other departments [16,24]. It is also noteworthy that most LOTE teachers are expected to teach a variety of courses ranging from disciplinary subject content (i.e., linguistics and literature) to language proficiency, regardless of their training and education. Many LOTE teachers also need to provide pastoral care for students. In short, they may work in conditions that are unfavorable for them to conduct research and improve their research output for their performance evaluation. They are vulnerable in terms of job security due to the current 'one size fits all' approach in performance assessment, and for this reason, it is important to conduct this inquiry to understand whether or not they are likely to have sustainable future careers in Chinese universities.

## 3. The Study

To address these issues, the study collected multiple data from different sources. First, we collected, either from open access or interviews, human resource documents or practice on personnel evaluation and promotion from 16 universities. Since universities of different standings will adopt different performance evaluation and staff promotion policies, we purposively sampled a variety of universities, including those classified as 'elite' universities (included in Scheme 985 and the Double First-class initiative) as well as regional universities in economically developed areas such as Beijing and Shanghai and hinterland areas such as Shaanxi and Ningxia. Table 1 presents information of the 16 universities whose policy documents were analyzed in the inquiry. As can be seen in the table, seven of the universities are 'elite' universities included in the government's Scheme 985 and the Double First-class initiative. The other nine universities are regional universities. Eight of the universities are located in China's hinterland provinces, while the remaining eight are based in economically developed coastal

regions. All the universities claim to place a strong emphasis on research when evaluating individual academics for contract renewal and promotion.

**Table 1.** List of 16 universities surveyed in the study.

| Number | Location Cities | Information of the Universities |
|---|---|---|
| **University 1** | Beijing | elite; in Scheme 985 and the Double First-class |
| **University 2** | Beijing | non-elite, regional university |
| **University 3** | Changchun | non-elite, regional university |
| **University 4** | Harbin | non-elite, regional university |
| **University 5** | Dalian | elite; in Scheme 985 and the Double First-class |
| **University 6** | Shanghai | elite; in Scheme 985 and the Double First-class |
| **University 7** | Shanghai | elite; in Scheme 985 and the Double First-class |
| **University 8** | Nanjing | elite; in Scheme 985 and the Double First-class |
| **University 9** | Nanjing | elite; in Scheme 985 and the Double First-class |
| **University 10** | Zhenjiang | non-elite, regional university |
| **University 11** | Hefei | non-elite, regional university |
| **University 12** | Hefei | non-elite, regional university |
| **University 13** | Chongqing | elite; in Scheme 985 and the Double First-class |
| **University 14** | Chongqing | non-elite, regional university |
| **University 15** | Xi'an | non-elite, regional university |
| **University 16** | Yinchuan | non-elite, regional university |

### 3.1. Research Informants

In the study we interviewed 32 purposively sampled informants (see Table 2); two from each of the schools or departments of foreign languages in the universities, concerning their understanding of performance review policies and their views on the sustainability of these policies with a focus on research productivity. All the informants were lecturers or newly promoted associate professors who were facing pressure for promotion, or who had just achieved promotion and were experiencing anxiety related to promotion to higher academic ranks (i.e., professors). As can be seen in Table 2, the participants had a variety of academic qualifications. It is noteworthy that 10 lecturers had Master of Arts qualifications, and all the newly promoted associate professors were PhD degree holders. This suggests that the 10 MA lecturers need to undertake PhD studies before they will be considered for promotion to associate professor in the universities in the inquiry. Nineteen of the lecturers were female and were of the age when they might be expected to fulfill many family responsibilities such as childcare.

**Table 2.** Information about the informants interviewed in the study.

| Informants | Number | Average Length of Service | Gender | Degree |
|---|---|---|---|---|
| Lecturers | 27 | 10.7 | 19 female/8 male | 17 PhD/10 MA |
| Associate professors | 5 | 10.4 | 3 female/2 male | 5 PhD/0 MA |

Notes: Most Chinese universities adopt the following academic ranks: Assistant lecturer, lecturer, associate professor, and full professor. Full professors are further divided into four levels.

### 3.2. Data Collection and Analysis

As mentioned earlier, we collected a variety of data. First, we collected the performance evaluation and staff promotion policies in the 16 universities with the help of our research informants. We asked the informants to copy the pages related to language teachers in their university documents and send them to us for analysis. Second, we conducted online semi-structured interviews with the 32 informants, because of the geographical spread of the universities. The interviews, which were conducted in Chinese, were guided by the performance review and promotion policies in the informants' universities (see the Appendix A). In our interviews we asked them how they felt about the relevant policies on performance evaluation and promotion, what challenges they perceived with regard to the application

of these policies to their own performance review and promotion, and what they had been doing to respond to the identified challenges. In doing so, we explored how they viewed the feasibility and sustainability of these performance review and promotion policies.

Third, we also collected information and data (e.g., the numbers of journals, issues, and published articles in 2019) from domestic and international publication outlets for academic subjects (e.g., language and linguistics) related to language teachers in 2019 (see Table 3), since all the universities value publications in journals indexed in databases such as CSSCI (China Social Sciences Citation Index), SSCI (Social Sciences Citation Index), and A&HCI (Arts and Humanities Citation Index). Since only publications indexed in these outlets are considered in performance review policies and can be used for promotion, we tried to assess how realistic and sustainable it was for LOTE teachers in Chinese universities to publish their research for performance review and promotion.

**Table 3.** Publication outlets for foreign language studies in 2019.

| Journal Categories | Total Number of Journals | Total Issues |
|---|---|---|
| Domestic (CSSCI) | 24 | 122 |
| International (SSCI and A&HCI) | 206 | 1067 |
| Total | 230 | 1189 |

Notes: Domestic outlets in this study include 20 indexed journals of foreign language studies and four of Chinese studies in which all language teachers can publish. CSSCI = China Social Sciences Citation Index; SSCI = Social Sciences Citation Index; A&HCI = Arts and Humanities Citation Index.

The data were analyzed using content analysis, guided by the research questions. Regarding the policy documents, we first identified the sections on research productivity that could be applied to academics in the schools and departments of foreign languages. Second, we identified the research productivity measures that were employed in these policies, before we differentiated these indicators according to factors such as the amount and type of research funding or the number of research publications in indexed journals (both domestic and international). Then we compared the identified information on research productivity across different universities, before we concluded the analysis with some shared themes.

In terms of the interviews with the informants, we first transcribed the interviews before they were analyzed through multiple rounds of reading to identify individual informants' feelings, perceived challenges, and strategic responses. In the first round of reading, we familiarized ourselves with each informant's experiences and views in relation to their institutional policies. Then, in the second reading we focused on their emotional responses to the policies, their experiences of the policies as they were applied to their own performance review and promotion, and their views on the sustainability of the relevant policies. In the third reading we compared these answers across different informants' interview data, in order to identify the common themes in the informants' engagement with the relevant university policies.

Further readings of the data helped us to corroborate the informants' feelings and perceptions with the findings that emerged from the analysis of journal-related information. For instance, one lecturer from a regional university made the following complaint about journal publication:

*It was extremely difficult to have a paper accepted for publication in CSSCI journals. It took me a long long time to get any response from the journal, and in the end I got rejected. (Lecturer 1, University 9)*

This informant experienced enormous difficulties in getting her paper published in journals that were recognized by her university in its performance review and promotion policies. This challenge was then checked, verified, and confirmed by our analysis of publication productivity in the CSSCI journals in related fields. By carrying out this confirmatory procedure we can speculate about how sustainable the relevant performance review and promotion policy is, and also whether LOTE teachers like this informant will be able to follow a sustainable future career in Chinese universities.

In order to further enhance the quality of our interpretation, the authors approached the analysis of a selection of data extracts independently, and then compared our interpretations to achieve a satisfactory inter-rater agreement (72%). We also sent out our interview transcripts and preliminary interpretations to the informants for checking. We did not receive any requests for correction.

## 4. Results

In this section, we report on the results that emerged from the analysis of our data regarding Chinese universities' expectations of LOTE teachers in junior academic ranks. We will first document the expectations that we found in our policy analysis, and then go on to describe what we learned from the informants and other data sources about the feasibility and sustainability of these expectations of LOTE teachers in junior ranks.

### 4.1. Chinese Universities' Expectations of Languages Other than English (LOTE) Teachers in Junior Ranks

As shown in Table 3, all the sampled universities in this inquiry placed significant emphasis on both publications and government-funded research projects as key performance indicators for research. The universities also stipulated explicit minimum requirements for promotion in terms of publications and funded research projects; these emerged as major challenges for LOTE teachers like our informants in pursuing a sustainable academic career.

Our analysis identified variations in the use of research performance indicators across the different universities in the study, since the universities have different academic standings and they are also located in areas with different economic and social development levels. Nevertheless, all the lecturers who want to be promoted to the rank of associate professor need to successfully apply for one funded research project at the ministerial or provincial level and publish at least two (and up to eight) publications in CSSCI indexed journals. In lieu of CSSCI indexed journal publications, LOTE teachers may publish their articles in international journals that are SCCI and A&HCI indexed. For some time, Chinese universities have been actively encouraging academics to publish in international journals, especially indexed journals, because of their pursuit of internationalization and favorable rankings in international research assessment exercises.

New policies over the last couple of years seem to have partially quenched this enthusiasm for publishing in international journals, since academics are now reminded to publish their research in China and for a Chinese readership [25]. Chinese academics in humanities and the social sciences are also urged to be aware of ideological correctness in their publications [5]. Nevertheless, despite these shifts that may be happening in policy discourses, it seems that publications in Chinese and international indexed journals are still critical in helping LOTE teachers to survive performance reviews and achieve promotion—in fact, the more publications, the better. Some leading universities, such as University 8 in this study, have also introduced fine distinctions between different indexed journal publications in performance reviews, and begun to highlight the significance of publications in journals that are in the first quartile of CSSCI, SSCI, and A&HCI journals. One article in a top-ranked journal is considered equivalent to 1.5 or two articles in other indexed journals in the performance review and promotion process.

This variation in the use of research performance indicators deserves further attention, as it results in different pressures experienced by LOTE teachers in universities of different standings. 'Elite' universities that are included in Scheme 985 and the Double First-class initiative have noticeably higher expectations for academics, including LOTE teachers. On average, LOTE teachers in Double First-class universities are expected to publish a minimum of 5.57 indexed journal articles for promotion, while those in other universities need to publish only 2.33 articles. It is important to note that many Double First-class universities in developed coastal areas have also introduced a system that allows these universities to terminate employment with individuals who fail to achieve promotion within a certain number of years (usually six to nine years). This means that LOTE teachers in these universities have a limited number of years to achieve the required minimum number of publications in order to

continue their employment. In the analysis, we also found that universities in major metropolitan or coastal regions have much higher expectations in terms of publications for LOTE teachers than do universities in Northeast or West China. As can be seen in Table 4, LOTE teachers in universities in Beijing and Shanghai need to publish at least seven indexed journal articles before they can submit their promotion application for associate professorship. Other institutions, such as University 16 in North China, require fewer than two indexed journal articles.

**Table 4.** Minimum promotion requirements for young foreign language teachers in Chinese universities.

| No. | Double First-Class (Yes/No) | Minimum Promotion Requirements | |
| --- | --- | --- | --- |
| | | Government-Funded Research Project | Publications |
| 1 | Yes | 1 ministerial or provincial | 7 in domestic or international journals (preferably CSSCI, SSCI, or A&HCI) |
| 2 | No | 1 ministerial or provincial | 3 CSSCIs, SSCIs, or A&HCIs |
| 3 | No | 1 ministerial or provincial | 3 CSSCIs, SSCIs, or A&HCIs |
| 4 | No | 1 bureau-level or above | 2 SSCIs or A&HCIs, or 3 CSSCIs |
| 5 | Yes | 1 ministerial or provincial | 3 SSCIs or A&HCIs, or 4 CSSCIs |
| 6 | Yes | 1 ministerial or provincial | 4 CSSCIs, SSCIs, or A&HCIs |
| 7 | Yes | 1 national or 2 ministerial/provincial | 8 CSSCIs, SSCIs, or A&HCIs |
| 8 | Yes | 1 ministerial or provincial | 8 CSSCI or more; 1 top-ranked CSSCI, SSCI, or A&HCI = 2 CSSCIs |
| 9 | Yes | 1 ministerial or provincial | 6 CSSCIs, SSCIs, or A&HCIs |
| 10 | No | 1 bureau-level or above | 2 CSSCIs |
| 11 | No | 1 ministerial or provincial | 3 CSSCIs |
| 12 | No | 1 ministerial or provincial | 3 CSSCIs |
| 13 | Yes | 1 ministerial or provincial | 3 CSSCIs; an academic monograph = 2 CSSCIs |
| 14 | No | 1 ministerial or provincial | 3 CSSCIs; 1 top-ranked CSSCI, SSCI, or A&HCI = 2 CSSCIs |
| 15 | No | 1 bureau-level or above | 5 publications (at least 1 CSSCI or above); 1 top-ranked CSSCI, SSCI, or A&HCI, or an academic monograph = 2 CSSCIs |
| 16 | No | 1 bureau-level or above | 3 publications (at least 1 CSSCI) |

Notes: Funded research projects are classified by the levels of the government authorities who fund the projects, including bureau/municipal, provincial/ministerial, and national levels.

### 4.2. The Realities of Academic Publication and Research Funding Applications

The analysis identified that the universities in the study used two key performance indicators to evaluate the research performance of academics, including the number of publications in indexed journals (i.e., CSSCI, SSCI, and A&HCI) and government-funded research projects. This motivated us to undertake further analysis of opportunities for publication and research funding. We found that LOTE teachers' efforts to publish in indexed journals and apply for research funding are undermined by significant challenges.

First, publication in indexed journals is a highly competitive and selective process. CSSCI journals that are likely to accept submissions from foreign language teachers in Chinese universities have a finite number of pages for print, and they are unable to deviate from this due to the tight control imposed by the National Press and Publication Administration Department. These journals also have English language teachers in Chinese universities as their major contributors and readers. Consequently, English language teachers may have an advantage in publishing their manuscripts in CSSCI journals in comparison with LOTE teachers, whose scholarship attracts a much smaller readership. For instance, Chinese universities had about 17,000 Japanese language teachers and 100,000 English language teachers in 2015 [26]. Sampling statistics in Wang and Wang's study [26] show that the number of articles in CSSCI journals per each Japanese language teachers was 0.0009 in 2014 and 0.0014 in 2015. In contrast, the number of articles in CSSCI journals per each English language teachers in 2014 and

2015 was 0.0118 and 0.012, which is eight to 12 times more. Our search found that Chinese universities currently employ around 200,000 English language teachers [27,28]. In this inquiry, over a third of English language teachers in the 16 universities were lecturers, who are pressurized for promotion to associate professorship to achieve job security. Based on this proportion, we estimate that around 70,000 English language teachers hold the rank of lecturer, and they have to compete with English language teachers of other academic ranks (e.g., associate and full professors) in publishing their manuscripts in CSSCI journals. If every English language teacher working in a Chinese university had submitted a manuscript to a CSSCI journal in a field related to foreign language and literature in 2019, around one in every 97.9 submissions would have been published, since the total number of publications in CSSCI journals is 2043. This is already a highly depressing figure for English language teachers but the prospects for LOTE teachers is even more bleak.

In addition, LOTE teachers may not have the linguistic skills to get their manuscripts written up and published in international journals such as SSCI and A&HCI journals, since the majority of these indexed journals require submissions in English [29]. In contrast, English language teachers in Chinese universities can develop the linguistic skills more easily to write and submit papers for SSCI and A&HCI journals. SSCI and A&HCI journals have a higher number of publications (9561). Though these journals also receive submissions from researchers in different parts of the world, they offer an additional, well recognized option for English language teachers in Chinese universities to publish their works and thus can potentially reduce the challenges facing language teachers who need to publish in indexed journals for performance review or promotion. Since both English and LOTE teachers usually only have six to nine years to publish a specific number of articles, as stipulated in their relevant universities' performance review and promotion policies, neither CSSCI nor international indexed journals offer an easy route for publication for LOTE teachers who need to publish. Nevertheless, it can be contended that LOTE teachers do have significant challenges in getting enough indexed journal articles published for research performance review and promotion.

Second, the competition for government-funded research funding is fiercely competitive for language teachers in Chinese universities. Our analysis concluded that there were around 560 ministerial and national level government-funded research projects per year over the last few years, including 260 from the National Social Sciences Foundation, 230 from the Ministry of Education, and 50 follow-up funding support opportunities from both the National Social Sciences Foundation and the Ministry of Education. In each province, language teachers, including LOTE teachers, can apply for about 40 projects at provincial level in philosophy and the social sciences. When competing with other language teachers with more senior academic ranks for around 600 government-funded project opportunities, LOTE teachers are clearly disadvantaged as they do not have a sufficient number of publications in well-respected journals to support their grant applications. Unfortunately, LOTE teachers are very likely to find themselves in a vicious cycle as their failures in publishing their works in the indexed journals lead to their lack of success in grant applications, which further undermines their research endeavor to get quality works published in indexed journals.

These contextual realities, together with high expectations from universities regarding university academics' research performance, constitute significant challenges for LOTE teachers in Chinese universities, causing them to feel highly stressed and motivating them to undertake a variety of additional efforts to pursue a sustainable career.

*4.3. Individual LOTE Teachers' Experiences*

In our analysis of our LOTE teacher informants' experiences, we identified three major themes related to research performance review and promotion, including perceived challenges, emotions, and strategic responses. These themes helped us to appreciate the sustainability of the relevant performance review system as experienced by the informants.

First, the informants reported a variety of challenges that constrained their endeavors to carry out research for publication. One informant complained about a lack of time due to her busy teaching and other administration duties:

*Many of us have to teach more than 8 h, and some of us even have to teach 16 h. Since we have to teach so much, I do not think that I have enough time for research. (Lecturer 2, University 11)*

We were not surprised to find that many LOTE teachers have heavy teaching loads and other duties, because many LOTE programs, especially those that were launched in the last few years, are understaffed [6]. Since a large number of the LOTE teachers in the inquiry only had Master's qualifications, they need time to acquire adequate research training. Even the informants with PhD degrees may need to have their knowledge updated, as Lecturer 3 mentioned:

*It is increasingly difficult to apply for research funding. I failed in every application. We have so many new theories and research methods in the field. I find it really difficult to keep up with what is happening in the field. (Lecturer 3, University 10)*

This is not a perception unique to LOTE teachers; many academics in Chinese universities and in other parts of the world may feel the same inadequacy. Other informants also reported repeated failures in getting their manuscripts accepted for publication in indexed journals, such as Lecturer 4:

*It is so difficult to get anything published in indexed journals. After I submitted my manuscript to a Chinese journal, I received no response, no update. I just waited and waited. After a long time, I was told that my manuscript was rejected. There was no feedback on my work, either. (Lecturer 4, University 15)*

Second, it may not be surprising to learn that most of the informants experienced negative emotions during this process. For instance, Lecturer 5 shared her strong feelings of regret and failure in her interview:

*I feel that I am a failure. I always feel stressed, but I achieve very little. At the end of every year, I feel totally lost. I regret that I have worked too slowly. It was relatively easy a few years ago, but it is so difficult for me to get anything published or apply for research funding. I truly regret that I did not take action then. Maybe I will be asked to leave my current teaching position. Who knows!*

*(Lecturer 5, University 7)*

The feelings captured by the extract above were echoed by most of the lecturers who were working hard to achieve promotion to associate professorship. We also detected similar feelings expressed by the informants who were already associate professors, although one of them did express relief because of his recent promotion. According to Lecturer 6 from University 12, he felt that he was very lucky to be promoted, and he felt he had "finally landed on the land" and did not have to worry about job security anymore.

Third, we noted that the informants undertook a variety of efforts to overcome their perceived challenges and deal with their emotions related to their future careers. Most of these efforts related to changing their attitudes towards research, as Lecturer 7 admitted:

*There is not much that I can do. I need to change my attitude towards research. If I work harder and spend more time, I do not think that it will be so difficult to meet the requirements for promotion. I have not adopted the right attitude, and I have not really persisted. I think that I need to be serious about the requirements and persist in doing the right thing to overcome these hurdles in my career development, before I waste too much time and I have to face the inevitable. (Lecturer 7, University 9)*

During our exchange, we thought about telling Lecturer 7 about what we had discovered from our analysis of indexed journals and the challenges associated with publishing in these journals. However, we did not do so; instead, we appreciated her commitment to changing her attitude and investing more time in research.

Like Lecturer 7, Lecturer 8 from University 9 also believed that he should work on publishing his manuscripts first before he could meet the other requirements for promotion:

> *I need to start writing journal articles. I also know that it is important for me to have government-funded research projects. However, I need to have publications before my grant applications will be favorably reviewed. The reviewers expect to see some relevant publications. If I cannot publish my manuscripts in indexed journals, I should start publishing them in other, less prestigious journals. I think that we should prepare ourselves for a long drawn-out process of accumulation. (Lecturer 8, University 9)*

From Lecturer 8, we sensed that his positive attitude had a solid grounding because he described a more detailed plan than Lecturer 7. He appeared to be willing to take a step-by-step approach, which seemed sensible in the current climate of fierce competition to publish manuscripts in indexed journals.

Other informants (five out of 32) had also attempted to publish their manuscripts in international journals, though most of them have made efforts to do so or they were attracted to do so. For this reason, most of them planned to invest more time in refreshing themselves with updated knowledge and skills for conducting research in their disciplinary areas.

However, they also admitted that many of their colleagues seemed to be running out of time, and they were being pressurized to achieve according to the policy expectations so that they could survive the performance review and promotion process. Nevertheless, despite all their efforts we still feel that it will be highly challenging, if not impossible, for them to meet expectations in research performance reviews for promotion, casting doubt on their sustainable career futures in Chinese universities.

## 5. Discussion and Conclusions

In this study we have analyzed performance review and promotion documents from 16 Chinese universities of different academic standings and in different geographical regions, with a focus on the stipulations regarding research performance. In doing so, we endeavored to find out what expectations Chinese universities have for language teachers, particularly LOTE teachers, whose sustainable career prospects are critical for maintaining LOTE education and sustaining multilingualism in Chinese universities.

We also collected a variety of data on the contextual realities of academic publishing for LOTE teachers and their personal experiences of this, analyzing indexed journals and government-funded projects as well as conducting interviews with the teachers themselves. Our findings from the journal and project analysis confirm that LOTE teachers face enormous challenges in having their manuscripts published in indexed journals and successfully applying for government-funded research projects. These findings corroborate the informants' actual experiences of academic publishing and research funding application, helping us to appreciate the widespread feelings of anxiety and stress among our LOTE informants.

In our view, the current research performance review system seems unsustainable. It has become necessary for policy makers and senior leaders in Chinese universities to explore alternative metrics to measure and assess LOTE teachers' research contributions (see [10,11]). Nevertheless, we were encouraged to see that our informants undertook various efforts to respond to these challenges. Even though we felt that some of the informants might not have realistic plans and sufficient time to overcome the challenges entailed in the research performance and promotion review, we were heartened to see that they were not defeated. Instead, they invested time in learning new knowledge and skills for research, while some of them started trying to publish their manuscripts in less prestigious

journals to gain experience of publishing and even attempted to submit manuscripts to international journals [30,31].

While the above-mentioned findings confirm the profound impact of managerialist practices on the professional lives of individual academics [5,12,13], they also demand further attention from university management and policy makers. If the Chinese government is committed to promoting the learning and teaching of multiple languages other than English in Chinese universities, university management need to consider the contextual realities that LOTE teachers face, and find ways to retain dedicated LOTE teachers and sustain multilingualism in Chinese universities. Reflecting on the findings, we suggest that university management and policy makers should take the following measures to support the sustainable career development of LOTE teachers in Chinese universities:

1. We believe that it is important to have separate teaching and teaching/research tracks for LOTE teachers. The participants complained about having long teaching hours and had little time for research as most LOTE programs have a very small number of LOTE teachers and require LOTE teachers to invest substantial efforts in program development [6]. Consequently, it will be strategically important for Chinese universities to retain LOTE teachers who specialize in teaching, so that they may dedicate themselves to the development and refinement of high quality LOTE programs. However, it is also important for the development and refinement of LOTE programs to be underpinned by a thorough understanding of LOTE learning and teaching, acquired from rigorous research. Therefore, LOTE teachers on a research/teaching track will be needed to provide the knowledge base for the design and development of LOTE programs. Only these LOTE teachers, rather than all LOTE teachers, will need to go through research performance reviews for promotion as stipulated by university policies. It is likely that some universities have already implemented a similar division among LOTE teachers, but we believe it is important for Chinese universities to make it clear that LOTE teachers in both tracks are essential to the provision of successful LOTE programs.

2. It will be desirable for university management to reconsider what counts as quality research in LOTE education. It is quite crucial for Chinese universities to value and accommodate diversity in research scholarship. Research on LOTE education may have a much smaller readership, and for this reason many CSSCI journals may be unwilling to publish papers in this area. It is important for university management to develop a list of journals focusing on LOTE education, which have a dedicated readership, and which are willing to publish LOTE research scholarship. We appreciate that this may go against universities' efforts to improve their ranking positions in international exercises, but if they are willing to adopt more inclusive journal databases such as Scopus, rather than SSCI and A&HCI which are much more selective in terms of journal coverage, they will encourage LOTE teachers to access more journals that are published in languages other than English, and which welcome submissions related to LOTE education [29,31]. These relevant publications could be still counted towards the universities' research performance in international ranking exercises. In addition, policy makers and university presidents in Chinese universities may consider exploring and adopting new metrics like their counterparts in Europe to ensure sustainability in academic career development for Chinese academics [10,11].

3. It is important for Chinese universities to appreciate the fact that it takes time for LOTE teachers to develop their capacity for research. For this reason, it is critical for Chinese universities to provide continuous professional development opportunities for LOTE teachers, encouraging them to update their research skills and disciplinary knowledge. This is one of the major themes that we found in our interviews with the informants, echoing the findings from many other studies on language teachers in Chinese universities [14,20–22]. LOTE teachers may need to be assisted in improving their knowledge about research methodology, developing feasible research ideas, and learning about academic publishing in Chinese and international journals. If they are properly supported, LOTE teachers will be able to publish studies containing findings that constitute significant knowledge contributions to the field of language education [32–35].

It must be noted that the research issue could be better approached with a nation-wide survey among LOTE teachers in Chinese universities. The interview participants' accounts may not capture the most recent policy changes in Chinese universities. Nevertheless, the analysis still has helped us to identify the profound problems surrounding managerialist practices related to research performance review for the career advancement of LOTE teachers in Chinese universities. We believe that many LOTE teachers in contexts other than Chinese universities may also go through very similar experiences, which are shared by university teachers in many contexts. We understand that it may not be easy for universities to completely excise managerialism as an underlying ideology in university management, but nevertheless we call on university management and policy makers to provide what is needed to support LOTE teachers' sustainable career development. Only by supporting the careers of these academics will Chinese universities and universities in other contexts be able to achieve sustainable multilingualism with LOTE teachers playing a key role. The same should also be done to support university teachers for sustainable careers globally (see [10,11]).

**Author Contributions:** Y.G.: Research design, data collection and analysis, writing up; H.S.: Drafting and writing up; M.B.: Data collection and analysis, writing up. All authors have read and agreed to the published version of the manuscript.

**Funding:** This research was supported by the Project of Humanities and Social Sciences, Ministry of Education, China (grant number 20YJC740012), the China Postdoctoral Science Foundation (grant number 2019M651453) and 2019 Postgraduate Education Reform Project of Tongji University (grant number ZD19040402).

**Acknowledgments:** Our gratitude goes to the teachers from the 16 universities who took time out of their busy schedules to conduct the interviews.

**Conflicts of Interest:** The authors declare no conflicts of interest.

## Appendix A

**Table A1.** Interview Schedule.

| | |
|---|---|
| 1. | Can you briefly introduce yourself? How do feel about working in the current university? |
| 2. | Can you share your recent experience of promotion or recent performance review (if relevant)? |
| 3. | What were expected to achieve for promotion or what kind of expectations were discussed in performance reviews for you? |
| 4. | What are the minimum requirements for promotion in terms of publications and funded research projects for promotion (associate professor) in your university? |
| 5. | How do you feel about current policies on academic performance evaluation and promotion? |
| 6. | Have the relevant policies been changing? How do feel about these changes or lack of changes? |
| 7. | What challenges, if any, have you perceived with regard to the policies on performance review and promotion? Why? |
| 8. | What have you been doing in response to these policies? Or what do you plan to do? |
| 9. | Any thoughts on the impact of these policies on your professional life? |

Note: questions are subject to modification in response to emerging situations during the interview process.

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
