# Peer review of "Sustainable Careers of Teachers of Languages Other than English (LOTEs) for Sustainable Multilingualism in Chinese Universities"

_sustainability, doi:10.3390/su12166396_

Round 1
Reviewer 1 Report
This manuscript examines the sustainability of career prospects of LOTE teachers in China, in the light of growing popularity of English as the language of instruction. The authors address the difficulties this sector of teachers face when trying to publish in indexed journals, and hence to raise funding, and ultimately to get promoted.
The topic is extremely interesting since these managerial practices, that base promotion prospects on individual merits, are becoming common elsewhere, spanning not only other countries but also many other areas of knowledge. The situation described in the manuscript is strikingly similar to the reality in my country, including pressure to publish and heavy teaching loads.
The introduction provides a good overview of the extension and effect of managerial practices, and provided evidences are, in general, right, but the nucleus of the analysis and discussion are based on doubtful data and questionable assumptions.
The major weakness of the paper relates, in my view, to the comparison with English teachers. These sections contain many assumptions that are not adequately supported. Could you please explain your reasons for the following statements, or the empirical data that support them?
l 284 – English language teachers in Chinese universities seem to find it easy to publish…
- 289 – If every English teacher working in a Chinese University had submitted a manuscript… (why 1? Why not to get the actual figure from official registers? How accurate id this 1/97.9?
l 287 – around 70000 English teachers hold the rank of lecturers… (and? Why do you contribute this data, if the calculation is done in reference to 200000 teachers?
- 298 – the actual number of manuscripts submitted to indexed journals is significantly lower than 1 per year…
l.313 – if a quarter of the language teachers have one project in progress… their success rate is much lower than 1/250.
L319 – the chance of being awarded a provincial/ ministerial project is very limited… (why? The application is also written in English, and can be hindered by their linguistic skills?)
Moreover, I don’t see how these calculations of the likelihood to be published support the distinction between English and LOTE teachers; the calculations refer to the number of English teachers (200000 – l. 285), but are extrapolated to English and LOTE teachers (l.299). No similar indices are provided only for LOTE teachers that allow for comparison.
I don’t know how to interpret the fact that LOTE teachers have started to publish in less prestigious journals and improving their skills? Given the pressures imposed by the system, will this increase the sustainability of their career prospects?
The paper is absolutely focused on LOTE teachers; that’s fine, since the description of the context and the analysis of the results refers to them. However, I would like to know the opinion of the authors on the transferability of these results to other segments of the population.
Some minor questions include:
There are some small details lacking from the Material & Methods section; i.e., how were the target journals selected from the selected databases? What was the structure and the content of the interview to the informants?
Paragraph 199 – 208 belongs to Methods or to Results?
Figure 1 is unnecessary, since it adds nothing to the information already presented in Table 4. If the reason is to compare graphically coastal universities and Northern or Western universities, then this layer of information should be added to the graphic (e.g. different colouring, or classification in the caption). Indicate in the caption the meaning of the (n) in parentheses.
Line 100: some Chinese characters
Table 2: the format is misleading, it seems to identify gender and degree (i.e., 17 out of 19 females hold a PhD).
Author Response
Please see the attachment of Response to Comments and Suggestions from Reviewer 1
Comments and Suggestions
This manuscript examines the sustainability of career prospects of LOTE teachers in China, in the light of growing popularity of English as the language of instruction. The authors address the difficulties this sector of teachers face when trying to publish in indexed journals, and hence to raise funding, and ultimately to get promoted.
The topic is extremely interesting since these managerial practices, that base promotion prospects on individual merits, are becoming common elsewhere, spanning not only other countries but also many other areas of knowledge. The situation described in the manuscript is strikingly similar to the reality in my country, including pressure to publish and heavy teaching loads.
The introduction provides a good overview of the extension and effect of managerial practices, and provided evidences are, in general, right, but the nucleus of the analysis and discussion are based on doubtful data and questionable assumptions.
The major weakness of the paper relates, in my view, to the comparison with English teachers. These sections contain many assumptions that are not adequately supported. Could you please explain your reasons for the following statements, or the empirical data that support them?
(1) L. 284 – English language teachers in Chinese universities seem to find it easy to publish…
289 – If every English teacher working in a Chinese University had submitted a manuscript… (why? Why not to get the actual figure from official registers? How accurate id this 1/97.9?
- 287 – around 70000 English teachers hold the rank of lecturers… (and? Why do you contribute this data, if the calculation is done in reference to 200000 teachers?
- 298 – the actual number of manuscripts submitted to indexed journals is significantly lower than 1 per year…
L.313 – if a quarter of the language teachers have one project in progress… their success rate is much lower than 1/250.
- 319 – the chance of being awarded a provincial/ ministerial project is very limited… (why? The application is also written in English, and can be hindered by their linguistic skills?)
Moreover, I don’t see how these calculations of the likelihood to be published support the distinction between English and LOTE teachers; the calculations refer to the number of English teachers (200000 – l. 285), but are extrapolated to English and LOTE teachers (l.299). No similar indices are provided only for LOTE teachers that allow for comparison.
Response: We appreciate your positive comments on the relevant writing. The feedback has helped us to realize that we may have been too eager to put a case for LOTE teachers and we may have lost our objectivity in evaluating the challenges that all the language teachers experience in Chinese universities. We must admit that the lack of reliable data on LOTE and English language teachers (who are not distinguished in any statistical reporting) presents a huge challenge for us. We have taken up this challenge with the utmost efforts. For instance, the challenge for publishing in CSSCI journals has been confirmed by a study conducted on Japanese language teachers, the largest group of LOTE teachers in China. CSSCI journals (their articles are recognized in research performance review for LOTE and English language teachers), publish 8-12 times fewer articles per teacher than that of English language teachers. It must be noted that English language teachers might also develop the linguistics skills more easily to publish in international journals in English than LOTE teachers. As for 70,000 English teachers holding the rank of lecturers, this is an estimation based on the calculations of the sampled universities in the inquiry. Language teachers holding the ranks of lecturers or below are most pressured for research performance review and job security. Once they are promoted to associate professor (one level higher than lecturers), their jobs are secure in most universities (otherwise, they may be kicked out). As for grant application, one can not make a new grant application if they have a grant in progress (which usually lasts 3 to 4 years), but we do not have sufficient data to prove this and for this reason, we remove the problematic statements here. As for the reduced chance of getting grants, this is again related to the challenge of publishing in CSSCI journals since grant applicants are evaluated in terms of prior publications (and other achievements).
(2) I don’t know how to interpret the fact that LOTE teachers have started to publish in less prestigious journals and improving their skills? Given the pressures imposed by the system, will this increase the sustainability of their career prospects?
Response: Many thanks for suggesting this. This is exactly what we would like to advocate for LOTE teachers. The system has to change to become more accommodating for LOTE teachers. We have made this clear in the concluding section(L.444-L.447). The other reviewer gives an excellent suggestion for us to read and refer to the new generation metrics advanced by European researchers.
(3) The paper is absolutely focused on LOTE teachers; that’s fine, since the description of the context and the analysis of the results refers to them. However, I would like to know the opinion of the authors on the transferability of these results to other segments of the population.
Response: This is a very important point. In essence, we believe that we need to problematize the managerialist practices in performance review. We need to be more accommodating for different kinds of contributions made by language teachers and university teachers. We will strengthen this argument in the concluding section (e.g. L.469, L.485).
Some minor questions include:
(4) There are some small details lacking from the Material & Methods section; i.e., how were the target journals selected from the selected databases? What was the structure and the content of the interview to the informants?
Response: we have provided the relevant details. The journals are listed by CSSCI, SSCI and AHCI according to different academic subjects. We focus those on subjects related to language teachers (language and linguistics etc.) (L.168, L.175). We include the interview schedule as an appendix.
(5) Paragraph 199 – 208 belongs to Methods or to Results?
Response: we have included the data sample to illustrate our analysis and make our analysis more transparent. After careful consideration, we feel that it is better for us to keep these in the methods.
(6) Figure 1 is unnecessary, since it adds nothing to the information already presented in Table 4. If the reason is to compare graphically coastal universities and Northern or Western universities, then this layer of information should be added to the graphic (e.g. different colouring, or classification in the caption). Indicate in the caption the meaning of the (n) in parentheses.
Response: We thank you for this suggestion. After careful consideration, we decided to remove Figure 1 (L.275).
(7) Line 100: some Chinese characters
Response: we have taken the suggestion and deleted the Chinese characters in brackets “創建一流大學和一流學科,or雙一流” which mean ‘first-class’ universities and academic disciplines (L.108).
(8) Table 2: the format is misleading, it seems to identify gender and degree (i.e., 17 out of 19 females hold a PhD).
Response: we have taken the suggestion and changed the format of table 2. (L.158)

Reviewer 2 Report
I found the paper relevant and valuable form the point of view of the research problem addressed. It is a vivid debate around the assessment of academic career.
I have some suggestions that could raise the quality of the paper and its impact.
The manuscript’s title contains the term ‘sustainable careers.’ However, the concept of sustainability has not been discussed in the literature review section. I suggest expanding the literature review on sustainability in academic career development and relating it with concepts such as evaluation, productivity, career paths, metrics, open science. European University Association has significant work (including position papers, white papers) on academic career assessment and development. CESAER has similar work https://www.cesaer.org/content/5-operations/2020/20200610-white-next-generation-metrics.pdf
https://www.cesaer.org/content/5-operations/2020/20200210-white-careers-of-early-stage-researchers.pdf
I would prefer to present the methodology of the study in the abstract explicitly.
Please check the in-text citations. Some of them appear as superscripts.
Line 124: Please give more details about the HR documents you have collected and analyzed. Were they evaluation forms? Human resources policy documents? Career plans? How did you get access to those documents? Were they subject to content analysis as well?
Line 129: Mission word. Information on the …
Line 130: Table 7 does not exist. Please replace it with Table 1.
Line 138: The participants in the interviews were purposefully selected? Please discuss this aspect.
Lines 164-166: What type of data have you collected concerning the journals?
Line 265: Which is the source of data in Figure 1?
The Discussion and Conclusion section needs to be consolidated with references to other research on the same or similar topics.
Does this research study have some limitations?
Author Response
Please see the attachment of Response to Comments and Suggestions from Reviewer 2
Comments and Suggestions
I found the paper relevant and valuable form the point of view of the research problem addressed. It is a vivid debate around the assessment of academic career.
(1) I have some suggestions that could raise the quality of the paper and its impact.
The manuscript’s title contains the term ‘sustainable careers.’ However, the concept of sustainability has not been discussed in the literature review section. I suggest expanding the literature review on sustainability in academic career development and relating it with concepts such as evaluation, productivity, career paths, metrics, open science. European University Association has significant work (including position papers, white papers) on academic career assessment and development. CESAER has similar work https://www.cesaer.org/content/5-operations/2020/20200610-white-next-generation-metrics.pdf
https://www.cesaer.org/content/5-operations/2020/20200210-white-careers-of-early-stage-researchers.pdf
Response: Many thanks for suggesting important resources. We have incorporated the ideas in the literature review. These are very new ideas to many universities as at present, most universities are still using the practice that this paper problematizes. We also include references to the two documents in the suggestions (in the concluding section). (L.74, L.470, L.519)
(2) I would prefer to present the methodology of the study in the abstract explicitly.
Response: Many thanks for your suggestion. We have presented the methodology more explicitly in the abstract.
(3) Please check the in-text citations. Some of them appear as superscripts.
Response: We apologize for the formatting issue. We have corrected them in the revised version.
(4) Line 124: Please give more details about the HR documents you have collected and analyzed. Were they evaluation forms? Human resources policy documents? Career plans? How did you get access to those documents? Were they subject to content analysis as well?
Response: Many thanks for raising these important issues. Unfortunately, the HR documents that are available for collection do not have evaluation forms, career plans. The relevant documents are part of the university policy and regulation for performance review and promotion (they just state the minimum requirements for colleagues to enter the race or competition for promotion). They were subject to content analysis.(L.132)
(5) Line 129: Missing word. Information on the …
Response: Thank you very much for your careful suggestion. We have added the preposition after “information”.(L.137)
(6) Line 130: Table 7 does not exist. Please replace it with Table 1.
Response: In this sentence, “As can be seen in the Table, 7 of the universities are ‘elite’ universities…” The table refers to table 1 which is mentioned in the previous sentence. 7 belongs to the latter part as in “7 of the universities.” (L.138-L.139)
(7) Line 138: The participants in the interviews were purposefully selected? Please discuss this aspect.
Response: the participants were purposefully selected as we wanted the informants who are concerned with promotion-related performance review and had recently gone through promotion process. We have explained this in the description. (L.146-L.147)
(8) Lines 164-166: What type of data have you collected concerning the journals?
Response: we have specified that we collected the number of journals, issues and published articles). (L.174-L.178)
(9) Line 265: Which is the source of data in Figure 1?
Response: As pointed out by the other reviewers, Figure 1 has the same information in Table 4. We have deleted the figure. (L.271)
(10) The Discussion and Conclusion section needs to be consolidated with references to other research on the same or similar topics.
Response: we have included the relevant references in the discussion and conclusion. In particular, we include the two crucial documents you suggested on new generation metrics and supporting early career researchers. (e.g. L.293-L.298)
(11) Does this research study have some limitations?
Response: we have mentioned the limitations briefly in the conclusion but we do feel that our analysis has achieved the same desired outcome. (L.532-L.534)

Reviewer 3 Report
The article reflects the sentiments of not only LOTE teachers but also academics working in universities world wide with the increased adoption of managerialism or new managerialism ideas and practices in university governance requiring research publications and research funding for language teachers indeed.
A question: Which result leads to this suggestion that university management and policy makers should take the following 442 measures to support the sustainable career development of LOTE teachers in Chinese universities: 443
44 1. We believe that it is important to have separate teaching and teaching/research tracks for 445 LOTE teachers
Author Response
Please see the attachment of Response to Comments and Suggestions from Reviewer 3
Comments and Suggestions
The article reflects the sentiments of not only LOTE teachers but also academics working in universities worldwide with the increased adoption of managerialism or new managerialism ideas and practices in university governance requiring research publications and research funding for language teachers indeed.
A question: Which result leads to this suggestion that university management and policy makers should take the following 442 measures to support the sustainable career development of LOTE teachers in Chinese universities: 443
44 1. We believe that it is important to have separate teaching and teaching/research tracks for 445 LOTE teachers
Response: Thank you. We have indicated explicitly what findings we draw on to propose our suggestions by referring to the findings on the participants’ complaint about long teaching hours and the need for their investment of efforts in program development.

Round 2
Reviewer 1 Report
Dear authors,
Thank for your detailed and well argued answers to all my suggestions.
Most of my concerns have been adequately addressed, and as a result I feel that the manuscript has improved significantly.
If I may add something, I would suggest going a bit deeper into the concept of sustainability, and also distinguishing actual numbers from estimated figures, in section 4.2.
Author Response
Response to Comments and Suggestions from Reviewer 1
Comments and Suggestions for Authors
Thank for your detailed and well argued answers to all my suggestions.
Most of my concerns have been adequately addressed, and as a result I feel that the manuscript has improved significantly.
If I may add something, I would suggest going a bit deeper into the concept of sustainability, and also distinguishing actual numbers from estimated figures
Response: Thank you very much for your positive comments and valuable suggestions which as in round 1 have guided us to improve the manuscript.
- We illustrated our understanding of the concept of sustainability by adding an explanation of the “notion of sustainable careers” (L.65-68), and further presented its relation to the inquiry in the context of sustaining multilingualism in Chinese universities (section 2, esp.L.127-137).
To distinguish actual numbers from estimated figures and avoid misunderstandings, we improved the presentation of the instance in L.302-308. Expressions like “An estimate of” (L.302) and “Wang & Wang estimate that” (L.304) have been modified. And it should be noted that the statistics in this section were provided to indicate the bleak prospects for LOTE teachers in Chinese universities.

Reviewer 2 Report
Dear authors,
Thank you for your efforts to improve the manuscript. I appreciate your detailed answers to my comments and suggestions.
The manuscript has been significantly improved. In my opinion, a clearer conceptualization of the sustainability in relation to the academic career is needed. It would be valuable to see your understading of sustainability in the context of the study you have conducted. Maybe it is the idea that the 'one size fits all' approach does not work because the impact factor does not reflect all the facets of the academic work. This would relate very well with one statement in the Conclusion section: It has become necessary for policy makers and senior leaders in Chinese universities to explore alternative metrics to measure and assess LOTE teachers’ research contributions (see [8,9]).
Best regards!
Author Response
Response to Comments and Suggestions from Reviewer 2
Comments and Suggestions for Authors
Thank you for your efforts to improve the manuscript. I appreciate your detailed answers to my comments and suggestions.
The manuscript has been significantly improved.
In my opinion, a clearer conceptualization of the sustainability in relation to the academic career is needed. It would be valuable to see your understanding of sustainability in the context of the study you have conducted. Maybe it is the idea that the 'one size fits all' approach does not work because the impact factor does not reflect all the facets of the academic work. This would relate very well with one statement in the Conclusion section: It has become necessary for policy makers and senior leaders in Chinese universities to explore alternative metrics to measure and assess LOTE teachers’ research contributions (see [8,9]).
Response: We really appreciate your positive comments and valuable suggestions which as in round 1 have guided us to improve the manuscript.
We adopted the point you suggested that the 'one size fits all' approach does not work because the impact factor does not reflect all the facets of the academic work and fully considered the realization of the concept of sustainability in this study. Therefore, we illustrated our understanding of the concept by adding an explanation of the “notion of sustainable careers” (L.65-68), and further presented its relation to the inquiry in the context of sustaining multilingualism in Chinese universities (section 2, esp.L.127-137). It indeed relates very well with our conclusion section. Thank you.
